# Effect of Food Matrix and Treatment Time on the Effectiveness of Grape Seed Extract as an Antilisterial Treatment in Fresh Produce

**DOI:** 10.3390/microorganisms11041029

**Published:** 2023-04-14

**Authors:** Anahita Ghorbani Tajani, Bledar Bisha

**Affiliations:** Department of Animal Science, University of Wyoming, Laramie, WY 82071, USA; aghorban@uwyo.edu

**Keywords:** *Listeria monocytogenes*, antimicrobials, grape seed extract, produce

## Abstract

Listeriosis outbreaks were associated with contaminated fruits and vegetables, including cantaloupe, apples, and celery. Grape seed extract (GSE) is a natural antimicrobial with potential for reducing *Listeria monocytogenes* contamination in food. This study assessed the effectiveness of GSE to reduce *L. monocytogenes* on fresh produce and the impact of food matrices on its antilisterial activity. GSE showed MIC values of 30–35 μg/mL against four *Listeria* strains used in this study. A total of 100 g portions of cantaloupe, apples, and celery were inoculated with *L. monocytogenes* and treated with 100–1000 μg/mL of GSE for 5 or 15 min. Results were analyzed using Rstudio and a Tukey’s test. Treated produce had significantly lower *L. monocytogenes* counts than the control samples (*p*-value < 0.05). The inhibition was significantly higher on apples and lowest on cantaloupe. Moreover, a 15 min treatment was found to be more effective than a 5 min treatment in reducing *L. monocytogenes* on all produce types. The reduction in *L. monocytogenes* levels varied between 0.61 and 2.5 log_10_ CFU reduction, depending on the treatment concentration, duration, and produce matrix. These findings suggest that GSE is an effective antilisterial treatment for fresh produce, with varying levels of effectiveness depending on the food matrix and treatment time.

## 1. Introduction

*Listeria monocytogenes* is a Gram-positive, facultative anaerobic bacterium that is widely distributed in the environment and is known to cause severe foodborne illnesses in humans, especially in immunocompromised individuals and pregnant women [1]. Controlling *L. monocytogenes* contamination and its growth on some types of produce are difficult undertaking due to a number of variables. The neutral or high pH may provide favorable conditions for *L. monocytogenes* growth. Furthermore, *L. monocytogenes* is capable of growing at refrigeration temperatures which is typically at many fresh produce items are stored [2]. Consequently, effective solutions for inhibiting the growth of *L. monocytogenes* in vegetable matrices are required which are effective under such storage conditions.

The level of *L. monocytogenes* contamination in produce can vary depending on a number of factors, such as the type of produce, growing conditions, and handling practices [3]. However, it is known that *L. monocytogenes* can be found in a variety of fresh produce, including apples, celery, and cantaloupe, as well as other types of fruits and vegetables [4]. The risk of contamination can be increased by factors such as poor sanitation, inadequate refrigeration, and cross-contamination during processing and handling [4]. Additionally, the survival and persistence of *L. monocytogenes* on fresh produce can be influenced by a multitude of factors. Some of the main factors include temperature, pH, moisture content, and the presence of other microorganisms [5,6,7]. Therefore, it is important to implement effective strategies for controlling the growth and survival of *L. monocytogenes* in produce to minimize the risk of foodborne illness.

Grape extracts were shown to possess antibacterial properties against various foodborne pathogens, including *L. monocytogenes*. In a study by Xu et al. (2015) [8], the antibacterial effect of grape pomace extracts against *L. monocytogenes* was evaluated, with results showing that the grape extracts had significant antibacterial activity against all four bacterial strains tested, with higher concentrations resulting in larger inhibition zones. In addition, a study by Veldhuizen et al. (2016) investigated the sublethal injury caused by grape seed extract and garlic extract to *L. monocytogenes*, with results indicating that both extracts caused a significant amount of sublethal injury, which was mainly caused by membrane damage [9]. These findings suggest that grape extracts may have potential as a natural antibacterial agent for controlling foodborne pathogens in fresh produce matrices such as apples, celery, and cantaloupe, which were associated with *L. monocytogenes* outbreaks. The effectiveness of grape extract treatment, however, may vary depending on the food matrix and treatment time [10,11,12], highlighting the need for further research in this area. Effective control measures for *L. monocytogenes* contamination in fresh produce are crucial to minimize the risk of foodborne illness, and natural antimicrobial treatments such as grape extracts could be a promising solution.

Listeriosis outbreaks associated with contaminated fresh produce were reported worldwide, including outbreaks linked to cantaloupe, apples, and celery [13]. In 2011, a *L. monocytogenes* outbreak linked to cantaloupes infected 147 people, resulting in 33 deaths and one miscarriage in the United States [14]. In 2014, an outbreak of listeriosis associated with caramel apples infected 35 people, resulting in 7 deaths, in the United States [6]. These outbreaks highlight the need for effective measures to control *L. monocytogenes* contamination in fresh produce.

Fresh produce matrices, such as apple, celery, and cantaloupe, can pose a particular challenge to antimicrobial treatment used to control *L. monocytogenes*. This is largely due to their neutral or high pH, which can support the growth of this bacterium, as well as the ability of *L. monocytogenes* to persist in produce matrices for extended periods [15]. Therefore, it is important to develop effective strategies for controlling the survival and growth of this foodborne bacterial pathogen in these products. The use of natural and effective antimicrobial treatments such as grape seed extract (GSE), which was shown to significantly reduce *L. monocytogenes* levels in fresh produce, could be one such strategy. Given the potential of natural antimicrobial treatments such as GSE to control *L. monocytogenes* in fresh produce [9], it is essential to investigate the optimal treatment conditions for different types of produce to ensure their effectiveness.

The main objective of our study was to Investigate the effect of GSE on reducing *L. monocytogenes* in different vegetable matrices, namely apple, celery, and cantaloupe, with varying treatment durations and GSE concentrations. The results of our study showed that GSE is a potent and natural antilisterial treatment for various types of fresh produce, and its effectiveness varies depending on the type of produce and the treatment time. Our findings are significant, as they suggest that GSE has the potential to be used as a natural and effective alternative to traditional antimicrobial treatments for fresh produce. However, the choice of treatment time and produce matrix are important factors to consider in order to optimize the effectiveness of GSE as an antilisterial treatment for fresh produce. These implications are crucial for the food industry, as they provide insights into the use of GSE for ensuring the safety and quality of fresh produce.

## 2. Materials and Methods

### 2.1. Preparation of L. monocytogenes Inoculum

A composite of four strains of *L. monocytogenes* was used: ATCC 15313, ATCC 19115, ATCC 13932, and ATCC 19114. The strains were cultured separately for 24 h at 37 °C in 50 mL of tryptic soy broth containing 0.6% yeast extract (TSBYE) (Lab M, Lancashire, UK) [15]. To obtain stationary phase cultures, the second subculture of each strain was incubated for an additional 24 h at 37 °C using the same medium and volume. Following incubation, 1 mL of the stationary phase culture of each strain was removed and serially diluted in brain heart infusion (BHI) broth (Thermo Fisher Scientific, Waltham, MA, USA) 9 mL blanks to prepare bacterial inocula. Microbial numbers were estimated by surface plating onto tryptic soy agar (TSA) immediately after serial dilutions were performed and before each inoculation procedure.

### 2.2. Preparation of Grape Seed Extract (GSE)

A commercial grape seed extract (Bulk Supplement, Henderson, Nevada, USA) was used as an antilisterial agent. A 25 mg/mL solution of GSE was prepared in a mixture of distilled water and 10% ethanol (vol/vol) to improve its solubility [16]. The solution was filter-sterilized using a 0.2 um pore size filter. To reduce the final ethanol concentration to 0.5% and to achieve different concentrations, the GSE solution was diluted in distilled water to obtain the following concentrations: 100, 200, 500, and 1000 μg/mL, which were selected based on 3 times the minimum inhibitory concentration (MIC).

### 2.3. Determination of Minimum Inhibitory Concentration (MIC) of GSE against L. monocytogenes

The MIC of GSE against *L. monocytogenes* strains ATCC 15313, ATCC 19115, ATCC 13932, and ATCC 19114 was determined using a broth microdilution method. The GSE stock solution was diluted in MH broth to concentrations ranging from 5 to 100 μg/mL. Triplicate wells were used for all treatments. *L. monocytogenes* strains were added to each well at an initial concentration to give 5 × 10^5^ CFU per well. The microplates were then incubated at 37 °C for 24 h. The growth of *L. monocytogenes* strains was monitored using the Spectramax M5 Microplate Reader (Molecular Devices, Sunnyvale, CA, USA) and the instrument’s wideband setting with a range of 420 to 580 nm [16].

### 2.4. Determination of Total Phenolic Content in Grape Seed Extract (GSE)

The total phenolic content of GSE was evaluated using the catechin reagent (Sigma-Aldrich, Co. LLC, Saint Louis, MO, USA) method described by Medina [17]. In this method, the phenolic content reacts with a diazonium salt in an alkaline environment to form a stable diazochromophore that can be measured at 420 nm. A 2% solution of GSE in water–ethanol was prepared and used to measure the concentration of phenolic compounds in the sample, expressed as the number of catechin equivalents (CE). CE is a measure of the concentration of phenolic compounds in a sample, which is determined by measuring the absorbance of the sample at 280 nm and comparing it to the absorbance of a standard solution of catechin. V is the volume of the sample used, DF is the dilution factor used for the sample, and W is the weight of the sample used. The overall phenolic content was calculated by multiplying the concentration of phenolic compounds (in catechin equivalents) by the volume or weight of the sample. Therefore, the formula used to calculate total phenolic content is Total phenolic content = (CE × V × DF)/W.

### 2.5. Preparation of Cantaloupe, Celery, and Apple Samples for Inoculation and GSE Dipping Treatment

Fresh cantaloupe, celery, and apples were purchased from a retail market and stored at room temperature until use. On the day of the experiment, 100 g portions of the rind of the cantaloupe, apple, and celery were cut using a sterilized stainless-steel knife. To obtain approximately 7 log CFU of *L. monocytogenes* per 100 g of produce, 100 μL of a *L. monocytogenes* suspension brain heart infusion (BHI) broth (Thermo Fisher Scientific, USA) was spot inoculated onto each vegetable piece. The inoculated produce pieces were then incubated at 37 °C for 24 h to allow listerial growth.

To evaluate the effect of GSE on the reduction in bacterial populations, 100 g piece of cantaloupe, celery, and apple was separately dipped in solutions of GSE in distilled water at concentrations of 100, 200, 500, and 1000 μg/mL. The control treatments were performed using distilled water only. The vegetable pieces were submerged in the GSE solutions for 5 or 15 min, inside a biosafety cabinet, to ensure sterility. After exposure, the samples were removed using a sterile stainless-steel spatula and drained on sterile absorbent paper.

### 2.6. Evaluation of L. monocytogenes Survival on Produce Samples

The survival of *L. monocytogenes* on each of the vegetable pieces after exposure to GSE solutions was determined. Ten pieces of each produce sample within each treatment concentration were analyzed, with *L. monocytogenes* counts performed in triplicate. A laboratory blender (Stomacher 400, Seward, Norfolk, UK) was used to homogenize 25 g of each sample in 225 mL of 0.1% peptone water for 2 min. Ten-fold serial dilutions of the homogenized samples were performed, and 0.1 mL of each dilution was plated on PALCAM agar and incubated at 37 °C for 24 h. The results were expressed as log CFU/g [15]. Additionally, to minimize residual effects of pH and added antimicrobial, the slurry was processed immediately for serial dilution and plated immediately.

### 2.7. Statistical Analysis

The data from the experiments performed in triplicate were analyzed using R Studio Version 1.4.1106 RStudio; https://www.rstudio.com (accessed on 10 August 2022). To identify significant variations among treatments in different vegetables and within each treatment concentration, Tukey’s test as a post hoc analysis and analysis of variance (ANOVA) were employed [18]. The results were considered significant if the *p*-value was less than 0.05.

## 3. Results

### 3.1. Determining the Total Phenolic Content in GSE

The result of the calculation showed that the concentration of catechin equivalents (CE) in the GSE used in the experiment was 8 nmol/μL. However, due to the interference caused by the natural pigmentation of GSE with the spectrophotometric assay, the total phenolic concentration in the sample could not be directly quantified. To determine the total phenolic content, a less-pigmented 2% solution of GSE was used. The equation used to calculate the overall phenolic content was CE = 8 nmol/μL ÷ (25 mg/mL x 1000 μL/mL), where 25 mg/mL was the concentration of GSE used to prepare the 2% solution, and 1000 μL/mL was the conversion factor for μL to mL. The calculation resulted in a value of 0.32 nmol/mg for the catechin equivalent per milligram of GSE.

### 3.2. Determining the Minimum Inhibitory Concentration (MIC) of GSE against Listeria spp. Strains

The results showed that GSE was able to inhibit the growth of all four *Listeria* strains at different concentrations. The MIC values of GSE against ATCC 15313, ATCC 19115, ATCC 13932, and ATCC 19114 were 30 μg/mL, 35 μg/mL, 30 μg/mL, and 30 μg/mL, respectively.

### 3.3. Investigating the Effect of Different GSE Solutions on L. monocytogenes Inactivation

The results of the experiment, which investigated the inactivation of *L. monocytogenes* on apple, celery, and cantaloupe after 5 min of treatment with different GSE solutions, are presented in Figure 1.

Immersing apples in a 100 μg/mL GSE solution for 5 min resulted in a significant decrease of 0.61 log CFU g^−1^ in the number of *L. monocytogenes* compared to immersing them in distilled water. With increasing concentrations of GSE to 200, 500, and 1000 μg/mL, the decrease in the number of *L. monocytogenes* was further enhanced to 0.7, 0.86, and 1.34 log CFU g^−1^, respectively. All of these reductions were significantly higher than the reduction observed with distilled water.

Dipping celery in 100 μg/mL GSE for 5 min reduced the number of *L. monocytogenes* by 0.75 log CFU/g, which is significantly higher than the reduction observed from dipping celery in distilled water for the same duration. The population of *L. monocytogenes* decreased significantly to 0.82, 0.95, and 1.43 log CFU/g when celery was immersed in 200, 500, and 1000 μg/mL GSE solutions, respectively.

When cantaloupe was dipped in 100 μg/mL GSE solution for 5 min, a significant decrease in the population of *L. monocytogenes* was observed compared to dipping in distilled water. However, a further significant decrease in the population of *L. monocytogenes* was not observed when the amount of GSE was increased to 200 μg/mL (0.6). A significant reduction of 0.65 and 1.0 log CFU/g was observed when the amount of GSE was increased to 500 and 1000 μg/mL, respectively.

The impact of GSE treatments with varying concentrations on *L. monocytogenes* inactivation on apple, celery, and cantaloupe after 15 min is illustrated in Figure 2.

The population of *L. monocytogenes* on apples, celery, and cantaloupe was significantly reduced after immersion in GSE solutions. For apples, a significant reduction of 0.75 log CFU/g was observed when immersed in a 100 μg/mL GSE solution for 15 min, which increased to 1.17, 1.41, and 1.64 log CFU/g when the concentration of GSE was increased to 200, 500, and 1000 μg/mL, respectively (*p*-value < 0.05). Similarly, dipping celery in a 100 μg/mL GSE solution for 15 min resulted in a significant reduction of 1.17 log CFU/g, which increased to 1.41, 1.64, and 1.81 log CFU/g when the concentration of GSE was increased to 200, 500, and 1000 μg/mL, respectively (*p*-value < 0.05). On the other hand, dipping cantaloupe in a 100 μg/mL GSE solution for 15 min resulted in a reduction of 0.59 log CFU/g, which was significantly higher than the reduction observed with distilled water. However, the reduction did not significantly increase when the concentration of GSE was increased to 200 μg/mL (0.6 log CFU/g). Nonetheless, a significant reduction of 0.6, 1.12, and 1.41 log CFU/g was observed when the concentration of GSE was increased to 500, and 1000 μg/mL, respectively (*p*-value < 0.05).

The statistical analysis showed that the main parameters affecting the reduction in *L. monocytogenes* populations on fresh produce treated with grape seed extract (GSE) are the concentration of GSE and the contact time. The results indicate that the higher the concentration of GSE and the longer the contact time, the greater the reduction in *L. monocytogenes* populations, with the time being the main parameter affecting the reduction.

### 3.4. Regression Analysis of L. monocytogenes Reduction with Different GSE Solutions, Fresh Produce Matrices, and Treatment Durations

The data presented in Figure 3 demonstrate the relationship between the concentration of GSE and the reduction in *L. monocytogenes* on apple, celery, and cantaloupe. The results show that there was a significant correlation between the increase in GSE concentration and the decrease in the bacterial population (*p* < 0.05). The analysis indicates that listerial reductions in different time periods and on different vegetables followed a linear pattern, with higher concentrations of GSE leading to greater reductions in bacterial populations. It can be observed that the grape seed extract treatments were associated with a good fit for the data, with all R-squared values greater than 0.5, indicating a relatively strong relationship between the grape seed extract treatments and the reduction in *L. monocytogenes* on the produce samples tested.

Specifically, the apple/15 min treatment had the highest R-squared value of 0.951, indicating that this treatment had the strongest relationship with the reduction in *L. monocytogenes* on apple after 15 min of exposure. The apple/5 min treatment also had a relatively high R-squared value of 0.848, indicating that this treatment was also effective in reducing *L. monocytogenes* on apple after 5 min of exposure.

The celery/5 min treatment had an R-squared value of 0.85, indicating a strong relationship with the reduction in *L. monocytogenes* on celery after 5 min of exposure, while the celery in 15 minutes’ treatment had a slightly lower R-squared value of 0.747.

Similarly, the cantaloupe in 5 minutes’ treatment had an R-squared value of 0.895, indicating a strong relationship with the reduction in *L. monocytogenes* on cantaloupe after 5 min of exposure, while the cantaloupe/15 min treatment had an R-squared value of 0.863.

## 4. Discussion

Grape and its byproducts were found to possess antimicrobial properties, making them potentially useful in the food safety. In particular, a study on wine pomace, a byproduct of grapes, demonstrated that both whole and separated skin pomaces from fermented (red) and unfermented (white) grape by-products exhibited antibacterial activity. The extracts were found to contain various phenolic compounds, including anthocyanins and flavanols, which may contribute to their antimicrobial activity [19]. Additionally, a study by Xu et al. [8] suggested that grape pomace could be a valuable source of natural bioactive compounds for the development of functional foods and nutraceuticals with health-promoting properties. Another study by Özkan et al. [20] found that grape pomace extracts from Emir and Kalecik karasi cultivars showed antibacterial activity against the majority of the 14 bacteria tested. These results suggest that grape byproducts could be a valuable source of natural antibacterial agents.

The antibacterial action of GSE polyphenols was shown to be effective against both Gram-negative and Gram-positive bacteria, both in laboratory settings and on fresh produce. Studies demonstrated the ability of GSE to inhibit the growth of bacteria such as *Escherichia coli*, *Salmonella typhimurium*, *Listeria monocytogenes*, and *Staphylococcus aureus* [21]. Additionally, GSE was shown to have antibacterial effects on fresh produce, including fruits and vegetables, with varying levels of effectiveness depending on the specific produce matrix and treatment time. The polyphenols in GSE are thought to be the primary component responsible for its antimicrobial properties, and their ability to disrupt bacterial cell membranes and inhibit essential enzyme systems was well documented in the literature [22]. The most abundant polyphenols in GSE are proanthocyanidins, which were identified as the main bioactive compounds responsible for the antimicrobial activity [23]. The antimicrobial properties of GSE are thought to be due to its ability to disrupt the cell membrane of bacteria, leading to cell lysis and death [24,25]. This mechanism of action is different from that of conventional antimicrobial agents, which typically target specific metabolic pathways or cell structures in bacteria [26].

To ensure bactericidal effect concentration of at least three times, MIC was used in all applied food inoculation studies.

GSE was shown to have antilisterial activity against the foodborne pathogen *L. monocytogenes* [27]. In this study, the effect of GSE on *Listeria* growth was investigated in different vegetable matrices, including apple, celery, and cantaloupe. The results showed that the antilisterial activity of GSE was influenced by both the concentration of GSE and the duration of treatment, with higher concentrations and longer treatment times leading to a greater reduction in *Listeria* growth. The study also found that the type of food matrix had an effect on the antilisterial activity of GSE, with cantaloupe being the most susceptible to *Listeria* growth, followed by celery and apple. The finding suggests that the effectiveness of GSE as an antilisterial treatment may be influenced by the texture of the surface of the food matrix, possibly due to differences in the ability of GSE to penetrate and interact with the bacterial cells on different surfaces. Further investigation into the impact of food matrix surface characteristics on the efficacy of GSE, as well as the underlying mechanisms, may provide valuable insights for optimizing the application of GSE as a food safety intervention.

A study of essential oils and their antimicrobial properties by Hyldgaard et al. also highlighted the importance of considering the food matrix in preservation strategies. The complex composition and structure of different foods can impact the effectiveness of antimicrobial agents, including essential oils. The interactions between food components and antimicrobials can lead to changes in the solubility, stability, and activity of the compounds, which can ultimately affect their ability to control microbial growth. Therefore, understanding the influence of food matrices on antimicrobial efficacy is crucial for the development of effective and safe preservation methods for different foods. This knowledge can inform the selection of appropriate antimicrobial agents and their optimal concentrations and treatment times to achieve the desired preservation outcomes [28].

Each type of vegetable matrix has its own unique physical and chemical properties, such as water content, pH, and nutrient composition [29]. These properties can influence the efficacy of GSE in inhibiting the growth of *Listeria* bacteria. Although differences in water content could theoretically affect the effectiveness of GSE treatment, it is unlikely to occur without an experimental setup that involves surface treatment. For example, the higher water content of cantaloupe and celery could dilute the GSE, reducing its effectiveness and potentially leading to less reduction in the population of *L. monocytogenes*. Additionally, the more acidic pH of apples compared to celery and cantaloupe could enhance the antimicrobial activity of GSE, resulting in a higher reduction in *L. monocytogenes* in apples. These factors can interact in complex ways, making it difficult to determine which factor has the greatest impact on the anti-listerial activity. Therefore, more research is needed to fully understand the mechanism of GSE and its effect on different produce matrices, as well as how these intrinsic factors can be optimized to enhance the anti-listerial activity.

Differences in texture and other natural barriers intrinsically present in different produce provide matrix-associated effects on antimicrobial treatments; however, the presence of naturally occurring antimicrobials in these foods might provide for even more strong interactions antimicrobial treatment, it is likely that the occurrence of natural antimicrobial in these foods might have an even greater effect on the application of the antimicrobial treatment. The presence of other antimicrobial compounds in produce samples can impact the interaction between the antimicrobial and *L. monocytogenes* [30]. Produce may contain natural compounds such as flavonoids, phenolic acids, and terpenes with antimicrobial properties that could enhance or hinder the effects of GSE. The effect of these compounds on the interaction with GSE varies depending on their type and concentration. For example, celery contains apigenin [31], a flavonoid that may contribute to the higher reduction of *L. monocytogenes* when combined with GSE. Similarly, cantaloupe contains carotenoids [32], which may also have antimicrobial properties but requires further investigation. Apples are another example of a food source that contains natural antimicrobial compounds. Quercetin, a flavonoid found in high concentrations in apples [33], possesses strong antibacterial properties and inhibits the growth of *L. monocytogenes*. Pectin, a polysaccharide found in apples, also has inhibitory effects against pathogenic bacteria [34], including *L. monocytogenes.* Malic acid, a major organic acid in apples, also demonstrated antimicrobial activity against *L. monocytogenes* [35]. However, the levels of these natural antimicrobial compounds in apples may vary, affecting the overall antimicrobial activity of GSE against *L. monocytogenes*. Further investigation is necessary to understand the specific compounds and their concentrations present in food sources and their impact on the antimicrobial activity of GSE. This knowledge would be valuable for optimizing the use of GSE as a natural antimicrobial agent in food industry applications.

To express antimicrobial effectiveness, a common method is the “CT value”, where C is the antimicrobial concentration (in ppm) and T is the contact time (in minutes). The CT value is a useful way to compare the sensitivity of different pathogens to a given antimicrobial treatment. However, for fresh-cut produce, other factors, such as the characteristics of the process wash water, must also be considered. For instance, whereas antimicrobials added to recreational waters may have minutes or longer to react with contaminants, the primary purpose of antimicrobials in produce wash water is to prevent cross-contamination, which can occur in a much faster, near instantaneous time frame. Therefore, the effectiveness of antimicrobials added to produce wash water is almost entirely dependent on concentration [36].

Post-harvest washing of fresh produce was a longstanding method to remove soil and debris and to reduce field-acquired contamination. However, despite the validation of washes by demonstrating a 5 log CFU reduction in relevant pathogens in laboratory settings, commercial conditions limit the actual log reduction to only 1–2, regardless of the sanitizer or washing time applied. This is due to factors such as internalization into the inner plant vascular system and organic loading within wash tanks, which contribute to the ineffectiveness of post-harvest washing. Additionally, the interaction of sanitizers such as chlorine with organic and inorganic components can form disinfection byproducts that neutralize antimicrobial action, leading to the dissemination of contamination between different batches. As a result, the industry shifted focus from decontaminating fresh produce to preventing cross-contamination. Recent studies evaluated the efficacy of antimicrobials in preventing cross-contamination during commercial post-harvest wash processes using classic culture techniques and advanced molecular characterization methods, such as whole-genome sequencing and metagenomics. These findings highlight the need for continued research and development of effective strategies to ensure the safety and quality of fresh produce [37].

## 5. Conclusions

This study showed that GSE treatment was able to significantly reduce the population of *L. monocytogenes* on apples, celery, and cantaloupe. The results also revealed that the effectiveness of GSE varied depending on the food matrix and treatment time, with the greatest reduction observed on apples and the lowest on cantaloupe.

These findings have important implications for food safety, as *L. monocytogenes* is a major public health concern and is often associated with the consumption of contaminated fresh produce. The use of GSE as a natural antimicrobial could potentially help prevent *L. monocytogenes* illnesses and improve the safety of fresh produce.

However, it should be noted that this study did not evaluate the survival of *L. monocytogenes* strains during cold storage. Further research is needed to investigate the long-term efficacy of GSE treatment in preventing the growth of *L. monocytogenes* during refrigerated storage, as this is an important consideration for the practical application of this treatment in the food industry. Additionally, further research is needed to investigate the impact of GSE on sensory qualities and potential interactions with other preservative methods in large-scale production. Nevertheless, the results of this study provide a promising foundation for the development of new natural antimicrobial treatments to ensure the safety and quality of fresh produce.

## Figures and Tables

**Figure 1 microorganisms-11-01029-f001:**
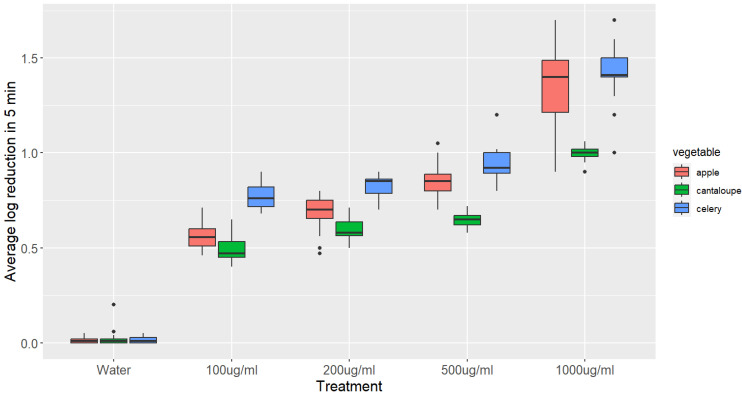
Boxplots showing the reduction in *L. monocytogenes* on different vegetables after 5 min treatment with various concentrations of GSE. The log reduction values were compared using an ANOVA, followed by a Tukey’s post hoc test. The boxplots represent the distribution of log reduction values for each treatment and vegetable, with the median, interquartile range, and range of values. The statistical analysis revealed that all of the log reductions were significantly higher than the control (*p*-value *<* 0.05), and the log reduction in the 1000 μg/mL treatment of apple was significantly higher than those in all other treatments (*p*-value *<* 0.05).

**Figure 2 microorganisms-11-01029-f002:**
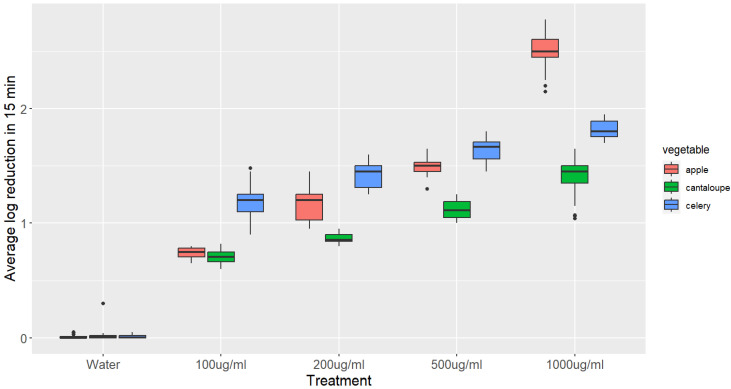
Boxplots showing the reduction of *L. monocytogenes* on different vegetables after 15 min treatment with various concentrations of GSE. The log reduction values were compared using an ANOVA, followed by a Tukey’s post hoc test. The boxplots represent the distribution of log reduction values for each treatment and vegetable, with the median, interquartile range, and range of values. The statistical analysis revealed that all of the log reductions were significantly higher than the control (*p*-value < 0.05), and the log reduction in the 1000 μg/mL treatment of apple was significantly higher than those in all other treatments (*p*-value < 0.05).

**Figure 3 microorganisms-11-01029-f003:**
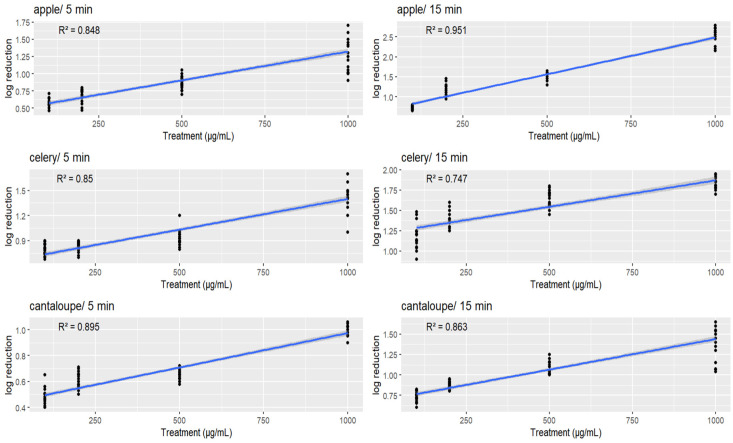
Scatter plots of log reduction in bacteria count on three different produce samples (apple, celery, and cantaloupe) after exposure to different levels of a treatment (in μg/mL) for either 5 or 15 min. Each data point represents a single experiment, and the blue line represents the linear regression model fit to the data. The R-squared value is shown on each plot, which indicates the proportion of the variability in the log reduction that is explained by the treatment level.

## Data Availability

The data presented in this study are available upon request from the corresponding author.

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
