# Peer review of "Effect of Food Matrix and Treatment Time on the Effectiveness of Grape Seed Extract as an Antilisterial Treatment in Fresh Produce"

_microorganisms, 2023, doi:10.3390/microorganisms11041029_

Round 1
Reviewer 1 Report
Grape Seed Extract (GSE) is a natural antimicrobial with potential for reducing Listeria monocytogenes contamination in food. This study evaluated the effectiveness of GSE to reduce L. monocytogenes on fresh produce and assessed the effect of food matrices on its antilisterial activity. 100g portions of cantaloupe rinds, apples, and celery were inoculated with L. monocytogenes and treated with 100-1000 ug/ml of GSE for 5 or 15 minutes. Results were analyzed using Rstudio and a Tukey's test (P<0.05) with analysis of variance. These findings suggest that GSE is a natural and effective antilisterial treatment for fresh produce, with varying effectiveness depending on the food matrix and treatment time.
this research is interesting and if possible, What is the effective active substance of grape seed extract? Please discuss in the discussion
Author Response
this research is interesting and if possible, What is the effective active substance of grape seed extract? Please discuss in the discussion
Thank you for taking the time to read our research and for your comment. We appreciate your interest in the active substance of grape seed extract. We have made sure we included the relevant information in our paper, specifically in lines 266 and 268 of the Discussion section. Please let us know if you have any further questions or concerns.
Reviewer 2 Report
A good effort that needs some improvements.
Line 20 – Add a line to introduce the bacterium.
Lines 21-22 - Something is missing here….and growth on some – Is this about the growth of LM? Add “its” before the growth
Line 22 – ….produce are difficult
Line 24- which is typically “at” many fresh produce items
Lines 29, 31, 36, 45, 51– Use L. monocytogenes
Line 31 - Do you have any idea of the level of LM contamination in produce?
Line 36-37 – Move the sentence from Ref 3 to the first para. Survival under refrigeration storage was mentioned there too.
The first and second paragraphs are overlapping a lot. Combine those.
Lines 46-49 – Don’t mention results under the introduction.
Line 61 – TSBYE should be moved after the yeast extract
Line 63 – Did you use the same medium and volume for the second sub-culture? How did you harvest cells? Did you measure OD? How do you know that 100 μl contains 7 logs?( Line 88)
Line 92 (section 2.5) – Are you talking about the inoculated and incubated produce? It is not clear that you used the inoculated produce. How many replicates were used per concentration of GSE?
Linea 101 & 104 – Triplicate plating overlaps
Line 106 – CFU/ cm2- Is this correct? Is that CFU/g?
Figure 1 – How did you get the log reduction? Did you check the counts of LM after inoculation (before and after incubation)?
Lines 146-147 – The higher the GSE concentration the lower the log reduction.
Figure 3 - I think it is better to add a table than to show the raw data here.
Line 200-201- Isn’t that the texture of the surface? (What do you mean by composition and structure)?)
Lin 208 - Did you measure the pH of each of the slurry you prepared using the stomacher?
Line 243-244 – What is this reference?
Author Response
Please see the attached revised version of the manuscript along with the point-by-point response to the reviewer's comments at the end of it. Apologies for any inconvenience.

Reviewer 3 Report
Manuscript 2281265
Journal Microorganisms
Title Effect of food matrix and treatment time on the effectiveness of grape seed extract as an antilisterial treatment in fresh produce
The manuscript entitled “Effect of food matrix and treatment time on the effectiveness of grape seed extract as an antilisterial treatment in fresh produce” describes the effect of GSE on Listeria monocytogenes inoculated on cantaloupe rinds, apple and celery. Different contact times were evaluated. The topic is not novel. The experimental plan lacks of the determination of the minimum inhibitory concentration of GSE against the four strains. The in vivo assays lacks of a cold storage period (Listeria is well adapted to low temperature). Moreover, the discussion of the results is limited. For these reasons, a major revision is strongly suggested. Please follow the comments in the file.

Round 2
Reviewer 3 Report
Manuscript 2281265
Journal Microorganisms
Title Effect of food matrix and treatment time on the effectiveness of grape seed extract as an antilisterial treatment in fresh produce
The manuscript entitled “Effect of food matrix and treatment time on the effectiveness of grape seed extract as an antilisterial treatment in fresh produce” describes the effect of GSE on Listeria monocytogenes inoculated on cantaloupe rinds, apple and celery. Different contact times were evaluated. The topic is not novel. The choice of the concentration tested is not clear. The in vivo assays lacks of a cold storage period (Listeria is well adapted to low temperature). Moreover, the discussion of the results is limited. For these reasons, a major revision is strongly suggested. Please follow the comments in the file.

Round 3
Reviewer 3 Report
Manuscript 2281265
Journal Microorganisms
Title Effect of food matrix and treatment time on the effectiveness of grape seed extract as an antilisterial treatment in fresh produce
The manuscript entitled “Effect of food matrix and treatment time on the effectiveness of grape seed extract as an antilisterial treatment in fresh produce” describes the effect of GSE on Listeria monocytogenes inoculated on cantaloupe rinds, apple and celery. Different contact times were evaluated. The topic is not novel. The organization of the manuscript should be revised. Moreover, the discussion of the results is limited. For these reasons, a major revision is still suggested. Please follow the comments in the file.

Round 4
Reviewer 3 Report
Auhtors revised the manuscript according to reviewer's comments. I have no further comments.
Author Response
Below is the document with authors answers
